# Clade 2.3.4.4b but not historical clade 1 HA replicating RNA vaccine protects against bovine H5N1 challenge in mice

David W. Hawman ®[1]✉, Thomas Tipih ®[1], Eddie Hodge ®[2], E. Taylor Stone[2], Nikole Warner ®[2], Natalie McCarthy[1], Brian Granger[2], Kimberly Meade-White[1], Shanna Leventhal[1], Kiara Hatzakis[2], Stephanie Park ®[2], Karen Gaffney[2], Kyle Rosenke ®[1], Jesse H. Erasmus ®[2]✉ & Heinz Feldmann ®[1]✉

The ongoing circulation of influenza A H5N1 in the United States has raised concerns of a pandemic caused by highly pathogenic avian influenza. Although the United States has stockpiled and is prepared to produce millions of vaccine doses to address an H5N1 pandemic, currently circulating H5N1 viruses contain multiple mutations within the immunodominant head domain of hemagglutinin (HA) compared to the antigens used in stockpiled vaccines. It is unclear if these stockpiled vaccines will need to be updated to match the contemporary H5N1 strains. Here we show that a replicating RNA vaccine expressing the HA of an H5N1 isolated from a US dairy cow confers complete protection against homologous lethal challenge in mice. A repRNA encoding the HA of a clade 1 H5 from 2004 (A/Vietnam/1203/2004) as utilized by some stockpiled vaccines, confers only partial protection. Our data highlight the utility of nucleic acid vaccines to be rapidly updated to match emergent viruses of concern while demonstrating that contemporary bovine H5N1 viruses can evade immunity elicited by historical HA antigens.

In early 2024, highly pathogenic avian influenza (HPAI) A H5N1 clade 2.3.4.4b was detected in United States dairy cattle. Since then, the virus has been found in dairy farms in 16 states[1–3]. In the United States, as of December 2024, 65 total human cases of H5N1 infection in humans had been reported, 64 occurring since March 2024[3]. Most human cases were associated with contact with poultry or cattle although one case in Missouri and another in California have unknown origins[3,4]. Though the current human public health threat posed by the US H5N1 outbreak is low, the wide circulation of H5N1 and continued spillover into humans could result in viral evolution that increases transmission to and among humans[5]. The United States has approved and stockpiled influenza A H5N1 vaccines based on the HAs of previously circulating H5N1 viruses[6–8] and could provide up to 125 million doses within four months[9]. However, the influenza A H5N1 currently circulating in cattle has acquired several mutations, including in the HA[10] an

immunodominant target of vaccine-elicited immunity[11], and it is possible that continued viral evolution will require updating of stockpiled vaccines.

Our replicating RNA (repRNA) vaccine is based on an alphavirus replicon system delivered by a cationic nanocarrier, called LION™[12]. Vaccines based on the LION/repRNA technology have received Emergency Use Authorization for SARS-CoV-2 in India[13] and has demonstrated preclinical efficacy against a variety of pathogens[12,14–16]. Here, we compared several repRNA vaccines in a recently developed lethal mouse challenge model[17] using a contemporary H5N1 isolated from a US dairy cow (Influenza A/bovine/OH/B24OSU-342/2024) hereafter referred to as A/bovine. We found that a repRNA vaccine expressing the HA of the historical clade 1 Vietnam H5N1 (A/Vietnam/1203/2004) (A/Vietnam), although immunogenic, provided poor protection against lethal A/bovine challenge.

[1]Laboratory of Virology, Division of Intramural Research, National Institute of Allergy and Infectious Diseases, National Institutes of Health, Rocky Mountain Laboratories, Hamilton, MT, USA. [2]HDT Bio, Seattle, WA, USA. ✉e-mail: david.hawman@hdt.bio; JesseErasmus@hdt.bio; feldmannh@niaid.nih.gov

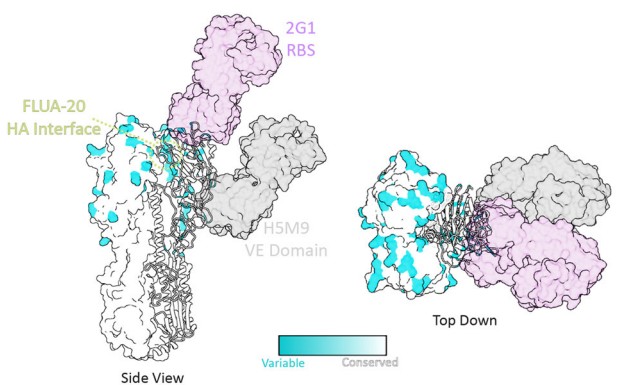

**Fig. 1 | Mutations within the A/bovine HA may impact neutralizing antibodies and vaccine efficacy.** The amino acid sequence homology between A/Vietnam and A/bovine was mapped onto the HA trimer structure (PDB 6CFG). Two of the protomers are displayed in the surface representation, and 1 protomer is displayed in ribbon representation. Cyan indicates regions with differences in sequence and areas colored white indicate conserved sequence homology. Fabs from well characterized cross-neutralizing antibodies (nAbs) that target three sites of vulnerability on the HA1 domain of the HA trimer are shown (PDB 4MHJ, 3GBN, and 4HG4 are aligned). nAb 2G1 targets the region including the receptor binding site (RBS) and is colored in purple, nAb H5M9 binds the VE domain (gray), and nAb FLUA-20 binds an occluded epitope on prefusion HA along the HA1 interface that can be exposed during reversible conformational changes (yellow).

In contrast, a single immunization with a vaccine expressing the HA of A/bovine conferred complete protection against challenge.

## Results

### The HA of A/Vietnam and A/bovine are antigenically distinct

To visualize the mutations in the HA of A/bovine, compared to the HA of the stockpiled H5N1 vaccines based on A/Vietnam, we mapped the sequence differences between the A/Vietnam and A/bovine HA to the HA trimer (Fig. 1). This analysis highlighted that most mutations between A/Vietnam and A/bovine HA were concentrated in the head domain of HA (Fig. 1). This region of the HA is targeted by neutralizing antibodies (nAb) and many of these mutations mapped to footprints of previously characterized neutralizing antibodies FLUA-20[18], 2G1[19], and HM59[20] that target three separate neutralizing epitopes in the HA (Fig. 1). This modeling suggests these mutations may impact nAb binding. As the HA head is an immunodominant target of vaccine-induced immunity[11] and nAbs are often the major correlate of vaccine-mediated protection against influenza, the sequence variation in the A/bovine HA could impact the protective capacity of vaccines using previously circulating H5N1 HAs. To test this hypothesis, we constructed repRNAs encoding the HAs of A/Vietnam (repHA-Vietnam) and A/bovine (repHA-Bovine) to understand how these differences would impact vaccine-mediated protection. We also evaluated a repRNA expressing the nucleoprotein of A/bovine (NP, repNP) alone or in combination with repHA-Bovine to determine if the inclusion of the more conserved NP antigen would increase protection. Cells transfected with these repRNAs confirmed appropriate high-level expression of both HA (Supplemental Fig. 1a, c) and NP (Supplemental Fig. 1b, c).

Next, C57BL6/J mice were vaccinated once intramuscularly with a cumulative dose of 10 μg of the repRNA. As a control, a cohort of mice received 10 μg of a repRNA expressing an irrelevant antigen (sham). Vaccination appeared well tolerated with no adverse events observed. Four weeks after vaccination, a cohort of mice was euthanized and serum analyzed for binding antibodies to recombinant HA from A/Vietnam and A/bovine by enzyme-linked immunosorbent assay (ELISA). Compared to the sham-vaccinated group, serum from mice vaccinated with either repHA-Vietnam or repHA-Bovine exhibited significant binding antibodies against both HAs (Fig. 2a). Interestingly,

we observed significantly reduced binding antibodies against heterologous HAs (Fig. 2a) suggesting that the mutations between the A/Vietnam and A/bovine HAs (Fig. 1) can significantly reduce binding of antibodies elicited by heterologous vaccination. We also measured binding antibodies to H5N1 strains from intermediate years between the 2004 A/Vietnam and 2024 A/bovine. Both repHA-Vietnam and repHA-Bovine elicited significant binding antibodies to HAs from A/duck/Laos/3295/2006 or A/bald eagle/Florida/W22-134-OP/2022 (Supplemental Fig. 2). Significant reactivity to NP was also measured in repNP-vaccinated mice (Fig. 2b).

RepHA-Vietnam vaccinated mice had low but detectable virus neutralization (VN) titers of 1:20 against homologous A/Vietnam but no detectable neutralization against heterologous A/bovine (Fig. 2c). RepHA-Bovine vaccinated mice had VN titers of 1:30 against homologous A/bovine but no detectable neutralization against heterologous A/Vietnam (Fig. 2c). Mice vaccinated with repNP + repHA despite having similar binding titers as mice vaccinated with repHA-Bovine alone, did not have detectable neutralizing activity (Fig. 2c) perhaps reflecting the lower dose of repHA-Bovine these mice received in the combination vaccination. We also measured neutralizing activity by hemagglutination-inhibition (HI) assay with authentic A/Vietnam or A/bovine. Using this assay, we did not measure any neutralizing activity against A/Vietnam or A/bovine (Supplemental Fig. 3). These data suggest that while repHA-Vietnam and repHA-Bovine elicit HA-binding antibodies after a single immunization, neutralizing activity is low or absent, may be dose-dependent and the mutations between A/Vietnam and A/bovine can impact nAb binding and neutralization activity. Serum from repNP-only vaccinated animals had no detectable neutralizing activity as expected (Fig. 2c). We also measured cellular immunity against the HA in vaccinated mice 2 weeks after prime vaccination by IFNγ enzyme-linked immunosorbent spot (ELISpot). Both repHA-Vietnam and repHA-Bovine elicited HA-specific T cells while repNP failed to elicit HA-specific T cells as expected (Fig. 2d). Cumulatively, these data indicate that our repRNA vaccines were immunogenic with repHA-Vietnam and repHA-Bovine eliciting both humoral and cellular immunity.

### repHA-Bovine vaccination protects against lethal A/bovine challenge

We have recently shown that WT C57BL/6J mice are highly susceptible to A/bovine challenge with mice rapidly developing lethal respiratory and neurological disease[17]. Therefore, 4 weeks after prime-vaccination vaccinated mice were challenged intranasally with 100,000 median tissue culture infectious dose (TCID$_{50}$) of A/bovine. Sham-vaccinated mice all succumbed with a mean-time-to-death (MTD) of 6.5 days (Fig. 2e, f). Surprisingly, four of six repHA-Vietnam mice also succumbed to the infection with a MTD of 8.5 days (Fig. 2e, f) while two mice exhibited no clinical disease following challenge (Fig. 2e, f), suggesting repHA-Vietnam provided incomplete protection against heterologous A/bovine challenge. In contrast, mice vaccinated with the homologous repHA-Bovine or repHA + repNP were completely protected from clinical disease (Fig. 2e, f). Despite NP-specific antibody responses in repNP-only vaccinated mice, these mice succumbed with similar kinetics as sham vaccinated mice (Fig. 2e, f), suggesting NP-specific responses were unable to protect against A/bovine challenge. Subsequent studies showed that the median lethal dose (LD$_{50}$) of A/bovine in C57BL6/J mice is <10 TCID$_{50}$s (Supplemental Fig. 4), indicating our challenge model represented a stringent challenge of vaccine efficacy.

### Vaccine-encoded H5 HA antigens elicit antibodies with distinct binding to homosubtypic HAs

The unexpected failure of the repHA-Vietnam to protect against heterologous A/bovine challenge warranted further investigation of the immunogenicity of repHA-Vietnam and repHA-Bovine. We developed a bead-based assay to evaluate antibody binding to H5 HAs from multiple clades of H5N1 viruses in parallel. We observed significant

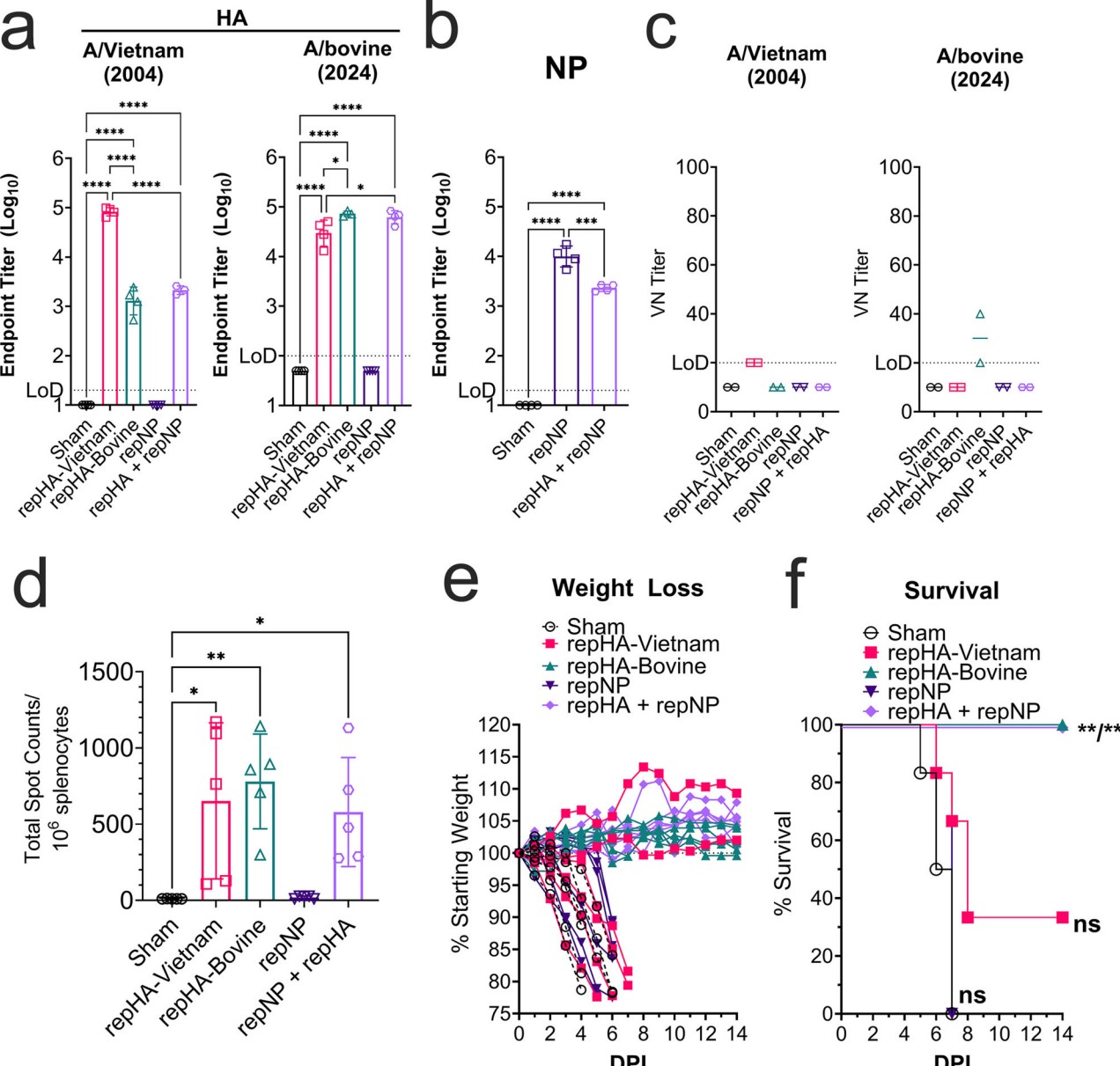

**Fig. 2 | RepHA-Bovine is immunogenic and protective against lethal A/bovine challenge in mice. a** Mice (*n* = 4 per group) were immunized with indicated repRNAs and HA-binding antibodies measured against indicated recombinant HAs (**a**) or NP (**b**). **c** Viral neutralization (VN) titers of pooled sera (*n* = 4 per group) were measured against infectious A/Vietnam or A/bovine virus and data shown as technical replicates. Splenocytes were evaluated for HA-specific T cells by IFNγ ELISpot. **d** Splenocytes were stimulated with overlapping peptides spanning the A/Vietnam HA in five pools and data shown as the cumulative spot count from all five peptide pools. **e**, **f** Mice (*n* = 6 per group) were challenged with 100,000 TCID$_{50}$

of A/bovine via the intranasal route and weighed daily (**d**) and euthanized when they reached >20% weight loss or exhibited signs such as severe lethargy or neurological signs. Box plots indicate mean plus standard deviation. **a**, **b** *P* values calculated with an ordinary one-way ANOVA with Tukey's multiple comparisons test. **d** *P* values calculated with an ordinary one-way ANOVA with Dunnett's multiple comparisons test to sham-vaccinated mice. **e** *P* values calculated with log-rank test with Bonferroni correction for multiple comparisons. \**P* < 0.05, \*\**P* < 0.01, \*\*\**P* < 0.001, \*\*\*\**P* < 0.0001.

differences in the binding profiles of antibodies elicited by each vaccination (Fig. 3a). Overall, repHA-Vietnam vaccination elicited antibodies that had binding activity to most H5 HAs but with a notable reduction in binding to A/bovine (Fig. 3a, b). In contrast, repHA-Bovine elicited antibodies that had strong binding to A/bovine, significantly greater than repHA-Vietnam vaccination, but with significantly reduced binding activity against many H5 clades compared to repHA-Vietnam vaccination (Fig. 3a, b). These data are consistent with our ELISA data and suggest that repHA-Vietnam and repHA-Bovine elicit antibodies with distinct homosubtypic reactivity. Interestingly, both repHA-Vietnam and repHA-Bovine vaccination failed to elicit

antibodies that bound clade 2.3.2.1 (A/Hong Kong 2007) HA suggesting that neither vaccine would provide optimal immunity against this virus in mice. We also evaluated pooled human sera from individuals receiving an investigational H5N1 vaccine based on A/Indonesia/05/2005 PR8-IBCDC-RG2 (clade 2.1.3)[21] and found that, like repHA-Vietnam vaccinated mice, human vaccinee sera had antibodies that bound most H5 HAs equally well (Fig. 3a). However, the human sera also reacted strongly to the A/bovine HA similar to our repHA-Bovine vaccinated mice (Fig. 3a) suggesting that pre-existing immunity, species-specific determinants, and/or the clade 2.1.3 HA utilized may improve the breadth of homosubtypic H5 HA reactivity.

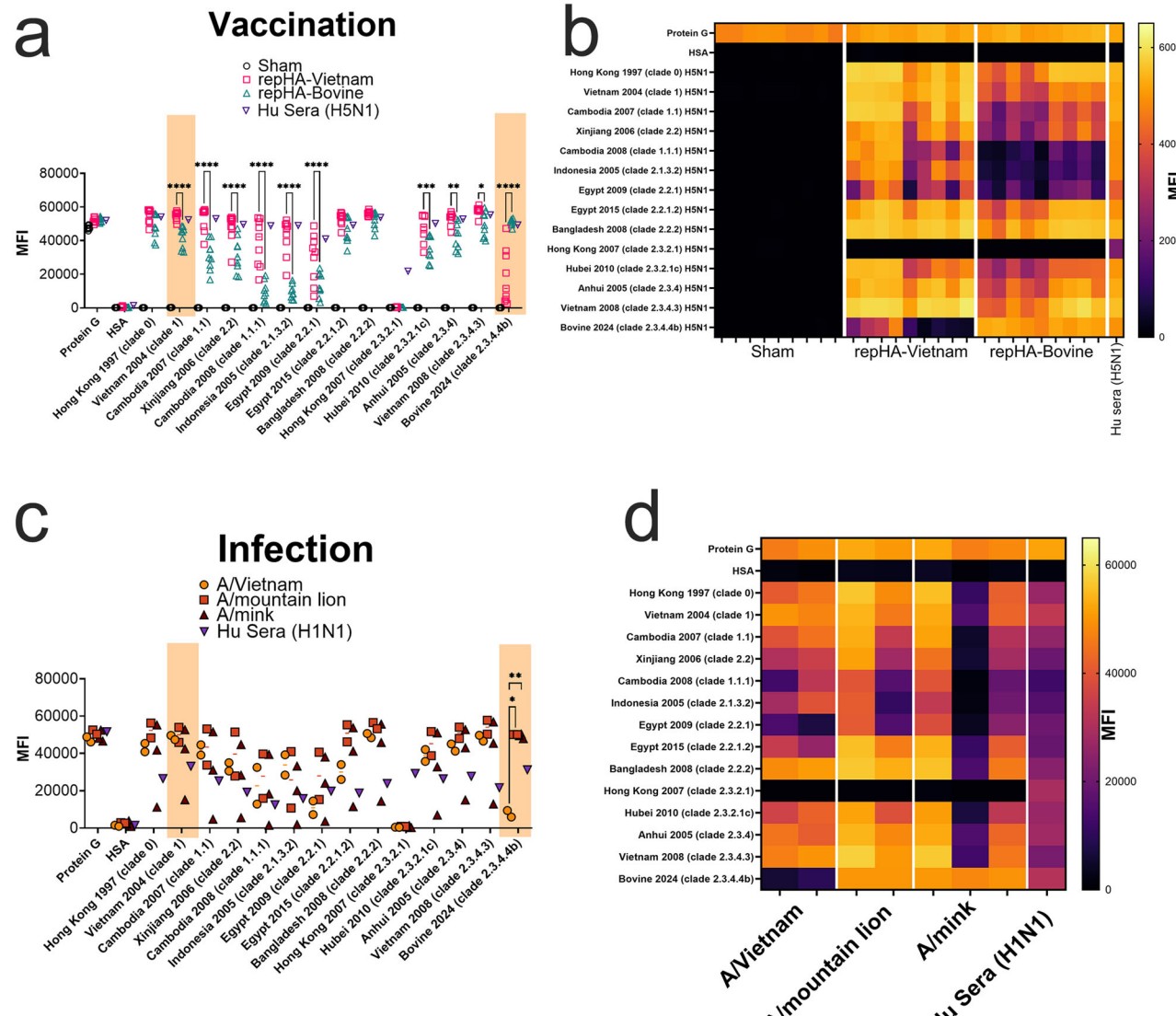

**Fig. 3 | RepHA-Vietnam and repHA-Bovine elicit antibodies with distinct homosubtypic reactivity.** Luminex microspheres were coated with the indicated recombinant HAs. H5 clade is indicated in parentheses. Antibody binding to recombinant HAs was measured in serum from sham or repHA-Vietnam or repHA-Bovine vaccinated mice (n = 9) (**a**, **b**). Data in (**b**) is duplicated from (**a**) but shown in heatmap format. N = 9 per group. As comparison, pooled human serum from humans receiving an experimental vaccine based on Influenza A/Indonesia/05/2005 (H5N1) was also evaluated. Data shown as median fluorescence intensity (MFI). Microspheres coated with protein G served as an internal positive control while microspheres coated with human serum albumin (HSA) served as an internal negative control. **c**, **d** Binding antibodies from serum from mice that survived

challenge with indicated viruses was evaluated as in (**a**) and data in (**d**) is duplicated from (**c**) but shown in heatmap format. As comparison, sera from a convalescent human infected with H1N1 in 2009 was included. **a**, **c** P values calculated with a two-way ANOVA with Tukey's multiple comparisons test. **a** Only comparisons between repHA-Vietnam and repHA-Bovine mouse sera with P values < 0.05 are shown for clarity. All comparisons between sham and repHA-Vietnam or repHA-Bovine vaccinated mouse sera was significant, P < 0.01 except for HSA and Hong Kong 2007 which were non-significant. **c** Only comparisons between mice and with P values of <0.05 are shown, all other comparisons were P > 0.05. *P < 0.05, **P < 0.01, ***P < 0.001, ****P < 0.0001.

To determine if these findings were unique to vaccination, we evaluated the binding profiles of antibodies elicited in C57BL6/J or BALBc/J mice that survived infection with A/Vietnam or recent clade 2.3.4.4b isolates A/mountain lion/MT/1/2024 (A/mountain lion) and A/mink/Spain/3691-8_22VIR10586-10/2022 (A/mink)[17,22]. We also evaluated human sera from a convalescent individual infected with influenza A H1N1 in 2009. No naïve mice challenged with A/bovine survived infection and therefore we were unable to evaluate sera from these mice. Mice that survived A/Vietnam challenge had antibodies with reduced binding to A/bovine HA, like vaccinated mice (Fig. 3c, d). In contrast, most mice that survived clade 2.3.4.4b infection had antibodies that bound both A/Vietnam and A/bovine HA (Fig. 3c, d) like repHA-Bovine vaccinated mice. However, one mouse that survived A/

mink infection had antibodies that largely bound only A/bovine HA suggesting infection can elicit antibodies that largely fail to react to other H5 HAs (Fig. 3d). Similar to our results with sera from vaccinated mice (Fig. 3a, b), mice that survived A/Vietnam, A/mountain lion or A/mink infection failed to produce antibodies that reacted to clade 2.3.2.1 HA (Fig. 3d) suggesting unique determinants of immunity within this HA. Convalescent sera from an individual infected with H1N1 had binding antibodies to most H5 HAs but with diminished reactivity compared to H5N1 infected mice (Fig. 3c, d) or H5N1 vaccinated humans (Supplemental Fig. 4). Cumulatively, these data suggest that the A/Vietnam and A/bovine HA antigens elicit antibodies either through vaccination or infection with distinct homosubtypic binding profiles possibly due to targeting of different epitopes within the HA.

We also evaluated heterosubtypic immunity to HAs from seasonal H3N2 and H1N1 influenza vaccine strains and other avian influenza A subtypes with documented human infections, H9N2, H6N1 and H10N8. Vaccination or infection of mice largely failed to elicit heterosubtypic binding antibodies with sera from most animals having little-to-no reactivity to H3, H1, H6, or H10 HAs (Supplemental Fig. 4). Human sera from H5N1 vaccinated individuals or a convalescent H1N1 infected individual had reactivity to most H3 and H1 HAs tested, likely through vaccination or exposure to seasonal influenza (Supplemental Fig. 4). These sera also had antibodies that bound H9, H6 and H10 HAs (Supplemental Fig. 4). Given the rarity of H9, H6 and H10 human infections[23–25] it is likely the binding antibodies represent heterosubtypic immunity elicited by vaccination or exposure to other influenza A subtypes.

## Discussion

Our data demonstrate that a single immunization with a repRNA expressing an HA from a contemporary H5N1 currently circulating within the United States confers complete protection against homologous lethal challenge in mice. Protection was associated with significant HA-binding antibodies that exhibited low-to-absent neutralizing activity against A/bovine. Surprisingly, heterologous protection was not observed in mice vaccinated with repHA-Vietnam suggesting bovine H5N1 can evade immunity elicited by historical HA antigens. Clinical evaluation of an updated H5 vaccine based on clade 2.3.4.4b A/Astrakhan/3212/2020 has recently begun[26]. A recent report evaluating an mRNA-based vaccine expressing the A/Astrakhan HA reported 3-fold lower neutralizing titers against A/pheasant/New York/22-009066-001/2022 that more closely matches the currently circulating H5N1 circulating in the United States[27]. Vaccination of chickens with recombinant viruses with clade 2.3.4.4 or 2.2.1.2 HAs showed strong homologous protection but incomplete heterologous protection[28] further suggesting that broad homosubtypic immunity may not always be achieved. Mice that survived A/mink infection, a clade 2.3.4.4b virus from 2022 had antibodies that bound A/bovine HA as strongly as mice that survived A/mountain lion infection, a clade 2.3.4.4b virus from 2024, suggesting that more recent clade 2.3.4.4b HA antigens such as A/Astrakhan may provide increased protection against A/bovine. It has recently been reported that a stockpiled H5N1 vaccine utilizing the clade 1A/Vietnam HA can elicit cross-neutralizing antibodies to clade 2.3.4.4b H5N1 in vaccinated humans[29], though the report did not evaluate contemporary 2024 H5N1 isolates. Humans receiving two doses of an investigational H5 vaccine based on clade 2.1.3A/Indonesia/05/2005 PR8-IBCDC-RG2 failed to elicit neutralizing antibodies against A/Vietnam or vice versa and instead heterologous prime-boost was necessary to elicit cross-neutralizing antibody responses[21].

We did not assess the impact on repRNA vaccination by pre-existing vaccine- or infection-induced immunity to influenza virus[30,31]. Humans with pre-existing influenza immunity vaccinated with an H5 vaccine developed greater broadly neutralizing antibody responses[11], suggesting pre-existing immunity may improve the breadth of response to H5N1 vaccination. Also, inactivated virus vaccine preparations such as the stockpiled Sanofi-Pasteur H5N1 vaccine[6] may elicit immune responses against other protective epitopes compared to our HA-only vaccine. Our evaluation of pooled human sera from individuals receiving an investigational H5 vaccine based on clade 2.1.3A/Indonesia/05/2005 PR8-IBCDC-RG2[21] showed broader H5 HA binding than vaccinated mice. These data support the hypothesis that pre-existing immunity to influenza may improve the breadth of responses elicited by H5 HA vaccines. Our evaluation of sera from mice surviving H5N1 infection suggests our results extend to immunity elicited by authentic viral infection arguing against altered expression of HA by our repHA vaccines driving unique immunity. A recent report evaluating the immunogenicity in mice of a lipid nanoparticle (LNP)

delivered replicating RNA expressing the HA of clade 2.3.4A/Anhui/1/2005 H5N1 reported low homologous neutralizing activity after a single immunization that was increased after boosting[32]. These data suggest boosting or higher vaccine doses may increase the neutralization capacity of repRNA-elicited antibodies. The repRNA/LION™ platform is less reactogenic than traditional LNPs suggesting higher tolerable single vaccine doses would be possible[33]. Cumulatively, our findings here suggest that currently circulating strains of H5N1 in the United States are sufficiently antigenically distinct to evade immunity elicited by historical HA antigens.

Our study has important limitations. Besides the previously discussed lack of pre-existing immunity in our mice, C57BL6/J mice challenged with A/bovine develop a severe neurological infection in addition to respiratory infection[17]. Distinct correlates of protection may be required to protect against neurological infection compared to protection from respiratory disease. Continued evaluation of these vaccines in non-human primates which develop a strictly lower respiratory disease course[17] is ongoing. Secondly, although we utilized a repRNA expressing the A/Vietnam HA as utilized by some stockpiled vaccines, we did not evaluate actual stockpiled vaccines that utilize distinct vaccine platforms. It is possible that stockpiled vaccines may elicit distinct immunity compared to repHA through presentation of the HA in more immunogenic conformations or inclusion of additional protective antigens and confer protection against heterologous H5N1 infection. In addition, the United States has approved H5N1 vaccines based on additional H5 HAs such as the Q-Pan H5N1 vaccine by GlaxoSmithKline which contains both A/Vietnam and the A/Indonesia/5/2005 H5N1 clade 2.1.3.2 HA[8,34] and our data showed that humans receiving an experimental vaccine based on the A/Indonesia HA had antibodies that bound to the A/bovine HA. However, the neutralizing capacity of these binding antibodies against A/bovine remains to be tested.

In summary, our findings highlight the utility of nucleic acid-based vaccines such as the repRNA platform to enable rapid updating of vaccines to match emerging viral variants. Our data show that a single immunization with a repRNA expressing the homologous HA confers complete protection against stringent lethal A/bovine challenge in mice. In contrast, a historical HA, as utilized by some stockpiled vaccines, conferred only partial protection, likely due to mutations in the HA head domain. A better understanding of the protection afforded by stockpiled H5N1 vaccines against the strains of H5N1 currently circulating in the United States is urgently needed.

## Methods

### Animals, biosafety, and ethics

All work with infectious influenza was performed in the maximum containment laboratory in accordance with standard operating procedures approved by the Rocky Mountain Laboratories Institutional Biosafety Committee, Division of Intramural Research, National Institute of Allergy and Infectious Diseases, National Institutes of Health (Hamilton, MT, USA). All animal work was performed in strict accordance with the recommendations described in the Guide for the Care and Use of Laboratory Animals of the Office of Animal Welfare, National Institutes of Health and the Animal Welfare Act of the US Department of Agriculture, in an AAALAC-accredited facility and study protocol approved by the RML ACUC. Male and female C57BL6/J mice 8 weeks of age were purchased directly from Jackson and randomly assigned to study groups. Animals were housed under controlled conditions of 30–35% humidity, 22 °C temperature and light (12-h light/12-h dark cycles). Food and water were provided ad libitum. All procedures on mice were performed under anesthesia. Human sera was obtained through BEI Resources, NIAID, NIH: Human Reference Antiserum to Influenza A/Indonesia/05/2005 (H5N1), Medium Titer, NR-33668[21] or Human Convalescent Serum 002 to 2009 H1N1 Influenza A Virus, NR-18965.

## Western blot and immunofluorescence

Baby hamster kidney-21 cells were transfected with in vitro transcribed repRNA. Twenty-four hours after transfection, for western blot lysates were collected and blots stained with mouse anti-HA (Genetex Catalog# GTX632358, Lot# 42317, 1:2500) or rabbit anti-NP (Genetex GTX125989, Lot#44902, 1:2500). For immunofluorescence, cells were fixed, permeabilized, stained with above antibodies and visualized with appropriate secondary antibodies conjugated to AlexaFluor594 (ThermoFisher, Cat#A1105, Lot#2897811 or Cat#A11012, Lot#2812001 both at 1:500). Hoechst was used to visualize nuclei.

## Vaccination and virus challenge

Codon-optimized gene sequences for the HA gene of A/Vietnam/1203/2004 or A/dairy cattle/Texas/24-008749-001-original/2024 (EPI_ISL_19014384, homologous to our stock of A/bovine) were synthesized and cloned into a plasmid vector encoding the 5' and 3' untranslated regions as well as the nonstructural open reading frame of Venezuelan equine encephalitis virus, strain TC-83, between PflFI and SacII sites, and clones sequence confirmed (Twist Biosciences). DNA was linearized by enzymatic digestion with NotI followed by enzymatic transcription and capping as previously described[35]. RNA was characterized by capillary gel electrophoresis to confirm production of full-length product and activity, including the production of the encoded protein, confirmed by western blot and immunofluorescence of cells transfected with each RNA. The vaccine was delivered intramuscularly via a single 50 µL injection. Sham-vaccinated animals were vaccinated with an equivalent cumulative dose of repRNAs expressing the glycoprotein precursor and nucleoprotein of the unrelated severe fever with thrombocytopenia syndrome virus. Influenza A/bovine/OH/B24OSU-342/2024 was propagated on Madin-Darby Canine Kidney (MDCK) cells. MDCK cells were from a long-term stock at Rocky Mountain Laboratories. The virus was grown in minimum essential media (MEM) supplemented with 4 µg/ml trypsin, 2 mM L-glutamine, 50 U/mL penicillin, 50 µg/ml streptomycin, 1× non-essential amino acids, 20 mM 4-(2-µculture infectious dose ($TCID_{50}$) on MDCK cells and sequenced to confirm identity and exclude biological contamination. Mice were challenged with 100,000 $TCID_{50}$ of virus diluted in Dulbecco's Modified Eagle Medium without additives via a 20 µL intranasal instillation for vaccine challenge study or indicated doses of virus in 50 µL DMEM for $LD_{50}$ studies. Serum from mice that survived H5N1 challenge was collected on day 28 post-infection.

## ELISA

Maxisorp (Nunc) or High-Bind (Corning) plates were coated with recombinant HA from A/Vietnam/1203/2004 (IBT Bioservices), A/dairy cow/Texas/24-008749-002 (SinoBiological), A/bald eagle/Florida/W22-134-OP/2022 (BEI) or A/duck/Laos/3295/2006 (BEI Resources) or recombinant influenza A NP (SinoBiological) at 100 ng/well in phosphate-buffered saline (PBS) overnight. Plates were blocked, and serial dilutions of mouse sera were applied. Bound antibody was detected with an anti-mouse IgG conjugated to horseradish peroxidase (Southern Biotech, Catalog #1030-05, Lot# D1922-YI62C, 1:4000 or Catalog #1031-05, Lot #H0021-QG93D, 1:10,000), plates developed with 3,3',5,5'-Tetramethylbenzidine (TMB) substrate (MilliporeSigma or SeraCare) and absorbance was read at 450 nM. Endpoints are reported as the reciprocal of the serum dilution to have an absorbance of 3 standard deviations above background absorbance as determined by curve-fit and interpolation of absorbance value.

## Neutralization assays

For the HI assay, sera was treated with RDEII (Seiken) overnight at 37 °C followed by heat inactivation at 56 °C for 1 h. Serially diluted sera was then mixed with 4 hemagglutination units (HAUs) of A/Vietnam or A/bovine for 1 h before incubation with turkey erythrocytes (Innovative Diagnostics) and evaluated for hemagglutination. The challenge virus was titrated in parallel to confirm challenge with 4 HAUs. For VN assay, 100 $TCID_{50}$ of virus was mixed with serial dilutions of heat-inactivated (56 °C for 30 min) sera for 1 h at 37 C. Residual infectious virus was then measured on MDCK cell monolayers. After 3 days, the supernatant was collected and presence of infectious virus determined by mixing with 0.33% turkey erythrocytes and measurement of hemagglutination. Titers are reported as the reciprocal of the last dilution to exhibit VN activity. Assays were performed on pooled serum ($n = 4$ mice per group) and measured in duplicate.

## ELISpot

Splenocytes were isolated from mice 14 days after prime vaccination by passing through a 70 µm cell strainer (Corning). MIAPS4510-Multiscreen plates (Millipore) were coated with rat anti-mouse IFN-gamma capture antibody (BD, Catalog #551216, Lot 4099272, used at 10 µg/mL) in PBS and incubated overnight at 4 °C. The plates were washed in PBS and then blocked (2 h, RT) with RPMI medium (Invitrogen) containing 10% heat-inactivated fetal calf serum (Gibco). Spleen cells were plated at $5 \times 10^5$ cells/well and stimulated with the following peptide pools from BEI: (NR-2609) Influenza Virus A (H5N1) Hemagglutinin (HA) Diverse Peptides and (NR-18974) Influenza Virus A/Vietnam/1203/2004 (H5N1) Hemagglutinin Protein. Peptides were reconstituted in 90% DMSO, 10% water at a concentration of 1 mg/mL and pooled in an approximately equimolar ratio where each pool contained 18–21 peptides. All stimulations were performed at a final concentration of 1.5 µg/ml/peptide and cultured for 20 h (37 °C, 5% $CO_2$). For negative controls, cells were stimulated with an equivalent concentration of DMSO. For positive controls, cells were stimulated with a final concentration of 20 ng/mL of Phorbol myristic acetate (PMA) and 1 µg/mL of Ionomycin. Biotinylated anti-mouse IFN-gamma antibody (BD, Catalog #554410, Lot# 3166827 or 1313950, Used at 1 µg/mL) and streptavidin-Alkaline Phosphotase-substrate (Biolegend, Catalog#34042, Lot#ZG395931, Used at 2 µg/mL) were used to detect IFN-gamma secreting cells. Spot-forming cells were enumerated using an Immunospot Analyzer from CTL Immunospot profession software (Cellular Technology Ltd).

## Bead-based multiplex HA-binding assay

Recombinant HA antigens (SinoBiological and IBT Bioservices) were individually coupled to xMap MagPlex-C microspheres (Luminex Corp.) at 10 ng per bead region using the Bio Plex Amine Coupling Kit (Bio-Rad), according to the manufacturer's instructions. The detailed strain name, provider and catalog numbers of each HA utilized are provided in Supplemental Table 1. Positive control microspheres were prepared by coupling 10 ng protein G (Thermo Fischer) to one bead region, and negative control microspheres were prepared by coupling 10 ng human serum albumin (Sigma Aldrich) to one bead region. All bead regions were combined and pipetted onto a 96-well plate at an estimated concentration of 500 beads per region per well. The plate was washed 2x with PBS, and mouse serum samples were applied at a dilution of 1:100 in staining buffer (Bio-Rad). Bound antibody was detected with a secondary anti-mouse (Southern Biotech Catalog #1038-08, Lot#C1717-NP53, used at 1 µg/mL) or anti-human IgG conjugated to biotin (Southern Biotech, Catalog #2014-08, Lot# C4221-P953E used at 1 µg/mL) at a concentration of 1 µg/mL diluted in staining buffer (Bio-Rad), then washed with PBS 3×. Streptavidin-PE (Southern Biotech, Catalog# 5105-09L, Lot# G0820-V504B, used at 2 µg/mL) was applied at 2 µg/mL diluted in staining buffer (Bio-Rad), and washed 2× with PBS before beads were resuspended in PBS-Tween. Analysis was performed using the MAGPIX System and the Luminex xPONENT for MAGPIX software. Results were reported as median fluorescence intensity (MFI), analyzing a minimum of 50 microspheres per region.

## Statistics

Indicated statistical comparisons were performed with GraphPad Prism v10.2.

**Reporting summary**

Further information on research design is available in the Nature Portfolio Reporting Summary linked to this article.

## Data availability

The data generated in this study are provided in the Supplementary Information. Source data are provided with this paper.

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

## Acknowledgements

The A/bovine virus was kindly provided by Richard Webby St. Jude's Children Hospital and Andrew Bowman Ohio State University. We also wish to thank Vincent Munster and Emmie De Witt and their laboratory

staff, Laboratory of Virology, NIAID for their efforts to obtain, propagate and titer virus stocks. This study was supported by the Intramural Research Program, NIAID. Funders had no input on study design, data interpretation or decision to publish.

## Author contributions

D.W.H., K.R., J.H.E., E.T.S., and H.F. designed and supervised the study. D.W.H., T.T., K.M.W., S.L., K.H., S.P., K.G., E.T.S., E.H., N.W., B.G., and N.M. performed experiments. D.W.H., J.H.E., E.T.S., N.W., S.P., E.H., and H.F. performed data analysis. H.F. and J.H.E. obtained funding. D.W.H. wrote the manuscript, and all authors agree to publication.

## Competing interests

J.H.E., E.T.S., N.W., S.P., K.H., E.H., and K.G. have an equity interest in HDT Bio. J.H.E. is an inventor on patents (US Patent nos. 11,458,209; 11,433,142; 11,752,218; 11,648,321; and 11,654,200) and patent applications (PCT/US22/76787, PCT/US23/60225, and PCT/US2024/010326) pertaining to the LION formulation and repRNA compositions described in the studies. Funders had no role in study design, data interpretation or decision to publish. The remaining authors declare no competing interests.
