## [Peer Review file · Nature Communications]

Clade 2.3.4.4b but not historical clade 1 HA replicating RNA vaccine protects against bovine H5N1 challenge in mice

Corresponding Author: Dr David Hawman

Version 0:

Reviewer comments:

Reviewer #1

(Remarks to the Author)

In their manuscript Hawman and colleagues test a replicating RNA vaccine against bovine H5N1 challenge in the mouse model. The manuscript is interesting and well done. However, there are several points that need the authors' attention.

Major points

- 1) The title needs to mention that this work was done in the mouse model.
- 2) How many LD50ies is the challenge dose?
- 3) In Figure 1A, please indicate the putative antigenic sites. Figure 1b and c could be deleted, they are not very useful.

Minor points

- 1) Line 27: Please update this number.
- 2) Line 45 and throughout the manuscript: Please use 'A/bovine' and not 'A/Bovine'. Currently both versions are used.
- 3) Line 68: Please define 'ELISA'.
- 4) Line 149: 'immunofluorescence', not 'Immunofluorescence'.
- 5) Line 149: Please define 'BHK-21'.
- 6) Line 150: Should be 'Western'.
- 7) Line 152: What are 'secondaries'? Please avoid lab slang.
- 8) Line 154-155: Please make sure this sentence is grammatically correct.
- 9) Please also define GPC, MEM, NEAA, HEPES, MDCK, DMEM, HRP, BEI (is BEI Resources meant?), PBS, TMB etc.
- 10) Line 169: Please use sentences, not sentence fragments.
- 11) Is there a reason why the sera were not RDE treated?
- 12) Line 177: Should be 'predictions', not 'Predictions'.
- 13) Line 186-187: Please correct the grammar of this sentence.
- 14) Figure 2: The colors used for sham and repNP look almost identical in a printed version. The colors should be changed to make them more distinguishable.

15) Suppl. Fig. 2: 'Duck' and 'Eagle' should be 'duck' and 'eagle'.

Reviewer #2

(Remarks to the Author)

General Comments :

Since the emergence of the H5 highly pathogenic avian influenza Gs/Gd lineage in 1996, it has occasionally infected humans, with a mortality rate exceeding 50%, and has garnered significant attention. Many countries have developed vaccines as a reserve to prepare for potential disease pandemics. Fortunately, for over the past 20 years, the disease has not been able to spread from person to person, posing no significant threat to human health. However, at the end of 2022, a virus from the H5 highly pathogenic avian influenza Gs/Gd lineage was discovered in a mink farm in Spain, which can spread between mammals, significantly increasing the likelihood of a global pandemic of H5 highly pathogenic avian influenza among human populations. Therefore, it is crucial to develop a human vaccine for H5 highly pathogenic avian influenza. This study utilized a replicating mRNA vaccine platform to prepare an H5 highly pathogenic avian influenza vaccine targeting hemagglutinin proteins from different viral strains (the national reserve vaccine strain VN1203 and the currently circulating viral strain). The immunogenicity of the vaccine was compared in mice, and the results showed that the vaccine derived from Clade 2.3.4.4b provided protection against the currently circulating strain, rather than the one from VN1203.

Specific Comments:

1. Since their first detection in domestic ducks in Eastern China (2013) and South Korea (2014), the H5Nx clade 2.3.4.4b viruses were isolated from wild birds in Qinghai Lake, China, in 2016 and spread on a large scale to Europe, Africa, Middle East, and Asia via migratory birds. The transatlantic spread to North America of H5N1 clade 2.3.4.4b HPAI viruses occurred in late 2021. Sequence data submitted to the GISAID Epi-Flu Database (<https://gisaid.org/about-us/acknowledgements/epiflu/>, accessed on 11 January 2024) demonstrates the global dominance of HPAI H5Nx viruses of clade 2.3.4.4b that have emerged over time [Animals (Basel). 2024 May 2;14(9):1372. doi: 10.3390/ani14091372.] (LINE1-2) . Therefore, it is unreasonable for this study to claim that the national reserve vaccine based on the VN1203 virus strain provides poor protection against the clade 2.3.4.4b viral strains solely based on the inadequate protective effect of the vaccine against the A/bovine/OH/B24OSU-342/2024 viral strain.

2. Replicating mRNA vaccines can induce a strong T cell response, and the assessment of T cell responses against HA proteins is an important indicator of vaccine efficacy. It is recommended to include information about this study in the article.

3. The study constructed replicating mRNA vaccines targeting HA and NP. To validate the expression level of the mRNA intracellularly, the researchers subjected the cell lysates to both non-denaturing and denaturing treatments, followed by analysis using Western blot; however, they did not demonstrate whether HA formed trimers(Supplemental Figure 1) .

4. The median lethal dose (LD50) of intranasal A/dairy cattle/New Mexico/A240920343-93/2024 in BALB/cJ mice was 30 plaque forming units 24 similar to A/bovine in C57BL6/J mice (Tipih et al. personal communication). Thus, our challenge dose of 100,000 TCID50 represents a stringent challenge likely in excess of 1000 LD50s (LINE125-128) . Before conducting the viral challenge experiments in mice, it is recommended to first determine the MLD50 (Median Lethal Dose) of the challenge viral strain.

5. It is recommended to include the preparation method of the replicating mRNA vaccine in the Materials and Methods section.

6. This work has moderate significance for the field and related areas. Some similar articles have already been published.

Immunogenicity and Cross-Protective Efficacy Induced by an Inactivated Recombinant Avian Influenza A/H5N1 (Clade 2.3.4.4b) Vaccine against Co-Circulating Influenza A/H5Nx Viruses.Mahmoud SH, et al. Vaccines (Basel). 2023. PMID: 37766075 Free PMC article.

Development of a nucleoside-modified mRNA vaccine against clade 2.3.4.4b H5 highly pathogenic avian influenza virus.Furey C, et al. Nat Commun. 2024. PMID: 38782954 Free PMC article.

Version 1:

Reviewer comments:

Reviewer #1

(Remarks to the Author)

In general, the authors addressed the reviewers' concerns well. However, not using RDE treatment because of one MDPI paper versus basically every protocol in the influenza world is not acceptable. RDE treated samples should be run and the results included in the manuscript.

Reviewer #2

(Remarks to the Author)

1. The study constructed replicating mRNA vaccines targeting HA and NP. To validate the expression level of the mRNA intracellularly, the researchers subjected the cell lysates to both non-denaturing and denaturing treatments, followed by analysis using Western blot; however, they did not demonstrate whether HA formed trimers (Supplemental Figure 1). Although we did not show production of trimers, our data suggests that repRNA vaccination presents the HA in relevant conformations to elicit protective immunity against a stringent lethal challenge. Further, we have now evaluated sera from vaccinated mice, vaccinated humans and mice that survived 2.3.4.4b infection against a panel of H5 HAs. We found that the antibodies elicited by repHA vaccination had similar binding profiles as those elicited by authentic A/Vietnam or clade 2.3.4.4b H5N1 viral infection. Thus, while we cannot be certain that repRNA vaccination results in production of authentic trimers, our data indicate that repRNA vaccination elicits antibodies of similar binding profiles as those elicited by authentic virus containing trimeric HA.

The authors still have not verified whether the HA form expressed by the self-replicating mRNA vaccine is in the trimeric form. The formation of HA trimers significantly enhances the induced immune response and protective effects. Therefore, I believe it is necessary to verify this using non-denaturing Western blot experiments.

REVIEWER COMMENTS

We thank the reviewers for their comments and suggestions to improve the manuscript. We have responded to the comments below. Line numbers refer to the clean resubmitted manuscript.

Reviewer #1 (Remarks to the Author):

In their manuscript Hawman and colleagues test a replicating RNA vaccine against bovine H5N1 challenge in the mouse model. The manuscript is interesting and well done. However, there are several points that need the authors' attention.

Major points

- 1) The title needs to mention that this work was done in the mouse model. We have added this clarification. Line 2
- 2) How many LD50ies is the challenge dose? In our first attempt we were unable to fully determine the LD50 of our bovine virus in C57BL/6J mice as even the lowest inoculating dose (10 TCID₅₀) was uniformly lethal. Future studies will evaluate lower doses of the virus but nevertheless, this indicates the LD50 of our virus is <10 TCID₅₀, below and new supplemental Figure 3.

- 3) In Figure 1A, please indicate the putative antigenic sites. Figure 1b and c could be deleted, they are

not very useful.

The putative antigenic sites for our vaccine are likely similar to other vaccines which elicit antibodies mainly to the head domain of the HA. This is also supported by our ELISA and bead-based binding assay which show that repHA-Vietnam and repHA-Bovine elicit antibodies with distinct binding activity against homologous and heterologous HAs. Since the mutations are concentrated in the head domain of the HA, therefore the most probable explanation is that the antibodies elicited by each vaccination target the HA head domain. We have also modeled well characterized neutralizing antibodies targeting three distinct epitopes to highlight that many mutations between A/bovine and A/Vietnam impact the binding footprints of the antibody. This may impact vaccine efficacy. We have increased the discussion of the hypothesis for figure 1, lines 48-58.

Figure 1b and c have been removed.

Minor points

- 1) Line 27: Please update this number. We have updated this to reflect most recent data on the CDC website as of October 23, 2024.
- 2) Line 45 and throughout the manuscript: Please use 'A/bovine' and not 'A/Bovine'. Currently both versions are used. Corrected throughout.
- 3) Line 68: Please define 'ELISA'. Added. Line 69
- 4) Line 149: 'immunofluorescence', not 'Immunofluorescence'. Corrected. Line 232
- 5) Line 149: Please define 'BHK-21'. Defined, Line 232
- 6) Line 150: Should be 'Western'. Corrected, Line 233.
- 7) Line 152: What are 'secondaries'? Please avoid lab slang. Clarified to secondary antibodies, Line 239.
- 8) Line 154-155: Please make sure this sentence is grammatically correct. Sentence removed and more comprehensive description of vaccine preparation provided. Line 238-249
- 9) Please also define GPC, MEM, NEAA, HEPES, MDCK, DMEM, HRP, BEI (is BEI Resources meant?), PBS, TMB etc. Added definitions where appropriate.
- 10) Line 169: Please use sentences, not sentence fragments. Corrected, line 264.
- 11) Is there a reason why the sera were not RDE treated? Reading through the literature, RDE treatment of sera was inconsistently used for VN assays and we therefore chose not to treat the sera as we found at

least one report that RDE treatment significantly reduced influenza specific IgG.
(<https://www.mdpi.com/2073-4468/12/2/39>).

12) Line 177: Should be 'predictions', not 'Predictions'. Structure prediction part of figure 1 and associated methods have been removed.

13) Line 186-187: Please correct the grammar of this sentence. Fixed, Line 304.

14) Figure 2: The colors used for sham and repNP look almost identical in a printed version. The colors should be changed to make them more distinguishable. We have improved the clarity of the figure. Sham is now represented by open circles with dashed lines for the weight loss figure. Throughout the rest of the figure we increased symbol size and changed them to open symbols to make it easier to see the overlapping symbols.

15) Suppl. Fig. 2: 'Duck' and 'Eagle' should be 'duck' and 'eagle'. Corrected.

Reviewer #2 (Remarks to the Author):

General Comments :

Since the emergence of the H5 highly pathogenic avian influenza Gs/Gd lineage in 1996, it has occasionally infected humans, with a mortality rate exceeding 50%, and has garnered significant attention. Many countries have developed vaccines as a reserve to prepare for potential disease pandemics. Fortunately, for over the past 20 years, the disease has not been able to spread from person to person, posing no significant threat to human health. However, at the end of 2022, a virus from the H5 highly pathogenic avian influenza Gs/Gd lineage was discovered in a mink farm in Spain, which can spread between mammals, significantly increasing the likelihood of a global pandemic of H5 highly pathogenic avian influenza among human populations. Therefore, it is crucial to develop a human vaccine for H5 highly pathogenic avian influenza. This study utilized a replicating mRNA vaccine platform to prepare an H5 highly pathogenic avian influenza vaccine targeting hemagglutinin proteins from different viral strains (the national reserve vaccine strain VN1203 and the currently circulating viral strain). The immunogenicity of the vaccine was compared in mice, and the results showed that the vaccine derived from Clade 2.3.4.4b provided protection against the currently circulating strain, rather than the one from VN1203.

Specific Comments:

1. Since their first detection in domestic ducks in Eastern China (2013) and South Korea (2014), the H5Nx clade 2.3.4.4b viruses were isolated from wild birds in Qinghai Lake, China, in 2016 and spread on a large scale to Europe, Africa, Middle East, and Asia via migratory birds. The transatlantic spread to North

America of H5N1 clade 2.3.4.4b HPAI viruses occurred in late 2021. Sequence data submitted to the GISAID Epi-Flu Database (<https://gisaid.org/about-us/acknowledgements/epiflu/>, accessed on 11 January 2024) demonstrates the global dominance of HPAI H5Nx viruses of clade 2.3.4.4b that have emerged over time [Animals (Basel). 2024 May 2;14(9):1372. doi: 10.3390/ani14091372.] (LINE1-2) . Therefore, it is unreasonable for this study to claim that the national reserve vaccine based on the VN1203 virus strain provides poor protection against the clade 2.3.4.4b viral strains solely based on the inadequate protective effect of the vaccine against the A/bovine/OH/B24OSU-342/2024 viral strain.

We believe our data to show that the contemporary North American 2.3.4.4b may be antigenically distinct compared to other H5N1 viruses. We have performed a more thorough measurement of reactivity of our vaccinated mouse sera against multiple H5s spanning multiple clades using a Luminex based multiplex assay (new Figure 3). We have found that mice vaccinated with the Vietnam HA elicit antibodies that cross react with multiple HAs but have a notable decrease in reactivity against clade 2.3.4.4b bovine HA by both our multiplex and ELISA. This does not appear to be unique to vaccination as a similar profile was observed in mice that survived A/Vietnam infection.

In contrast, mice vaccinated with the bovine HA elicit antibodies that strongly react to the clade 2.3.4.4b bovine but have reduced activity against several other clades suggesting the bovine HA elicits antibodies with reduced cross-reactivity but greater homologous reactivity. We believe this reactivity data and vaccine efficacy data has important implications for vaccines against H5NX viruses as first it suggests that the bovine HA is sufficiently antigenically distinct to escape antibodies elicited by the Vietnam HA. Secondly, our data also suggest that the H5 HAs can elicit antibodies with unique reactivity profiles. This could have implications for vaccine design and efficacy. We have added this as new figure 3, results line 108-155 and discussion lines 182-191.

However, we do agree that a significant limitation of this data is that these are mice lacking pre-existing immunity and may not accurately model vaccine efficacy in humans. We have also used this assay to evaluate sera from humans vaccinated with an H5N1 vaccine A/Indonesia/05/2005 PR8-IBCDC-RG2 (H5N1) (clade 2.1.3) (BEI Resources) or recovered from an H1N1 infection in 2009 (BEI Resources). Humans receiving the experimental H5 vaccine had reactivity to most H5 HAs including the A/bovine. This suggests that either the clade 2.1.3 HA used in that vaccine, species-specific determinants and/or pre-existing immunity improve the breadth of vaccine elicited reactivity. We have added a discussion of these limitations and others of our studies, lines 199-211. Cumulatively we feel that our conclusion that our data warrant evaluation of stockpiled vaccines against contemporary 2.3.4.4b H5N1 circulating in the United States is still supported by the data.

2. Replicating mRNA vaccines can induce a strong T cell response, and the assessment of T cell responses against HA proteins is an important indicator of vaccine efficacy. It is recommended to include information about this study in the article.

We have added the T-cell response data to figure 2 and results line 89-91.

3. The study constructed replicating mRNA vaccines targeting HA and NP. To validate the expression level of the mRNA intracellularly, the researchers subjected the cell lysates to both non-denaturing and denaturing treatments, followed by analysis using Western blot; however, they did not demonstrate whether HA formed trimers (Supplemental Figure 1) . Although we did not show production of trimers, our data suggests that repRNA vaccination presents the HA in relevant conformations to elicit protective immunity against a stringent lethal challenge. Further, we have now evaluated sera from vaccinated mice, vaccinated humans and mice that survived 2.3.4.4b infection against a panel of H5 HAs. We found that the antibodies elicited by repHA vaccination had similar binding profiles as those elicited by authentic A/Vietnam or clade 2.3.4.4b H5N1 viral infection. Thus, while we cannot be certain that repRNA vaccination results in production of authentic trimers, our data indicate that repRNA vaccination elicits antibodies of similar binding profiles as those elicited by authentic virus containing trimeric HA.

4. The median lethal dose (LD50) of intranasal A/dairy cattle/New Mexico/A240920343-93/2024 in BALB/cJ mice was 30 plaque forming units 24 similar to A/bovine in C57BL6/J mice (Tipih et al. personal communication). Thus, our challenge dose of 100,000 TCID50 represents a stringent challenge likely in excess of 1000 LD50s (LINE125-128) . Before conducting the viral challenge experiments in mice, it is recommended to first determine the MLD50 (Median Lethal Dose) of the challenge viral strain.

To address the public health urgency presented by A/bovine we chose to proceed with viral challenge before the LD50 for A/bovine had been fully established. We do now know that the LD50 of A/bovine in C57BL6/J mice is <10 TCID50. We have included these data in supplemental figure 3.

5. It is recommended to include the preparation method of the replicating mRNA vaccine in the Materials and Methods section.

Added. Lines 238-249

6. This work has moderate significance for the field and related areas. Some similar articles have already been published.

Immunogenicity and Cross-Protective Efficacy Induced by an Inactivated Recombinant Avian Influenza A/H5N1 (Clade 2.3.4.4b) Vaccine against Co-Circulating Influenza A/H5Nx Viruses. Mahmoud SH, et al. Vaccines (Basel). 2023. PMID: 37766075 Free PMC article.

Development of a nucleoside-modified mRNA vaccine against clade 2.3.4.4b H5 highly pathogenic avian influenza virus. Furey C, et al. Nat Commun. 2024. PMID: 38782954 Free PMC article.

We have referenced Furey et al. and believe their data may suggest similar as their vaccine is based on A/Astrakhan from 2020 and already saw 3-fold reduced neutralizing activity against a 2022 2.3.4.4b virus. We have included discussion of the Mahmoud et al. paper (Line 166-168) and added discussion of Blishe et al (<https://academic.oup.com/jid/article/203/5/666/894361?login=true>) (Line 174-176 and 185-188) which also shows that homologous vaccination with H5 HAs failed to elicit heterologous strain

neutralizing antibodies. To our knowledge, our manuscript is the first to report efficacy testing of vaccines against a contemporary 2024 2.3.4.4b isolate of the kind currently posing a public health threat in the United States.

Reviewer #1 (Remarks to the Author):

In general, the authors addressed the reviewers' concerns well. However, not using RDE treatment because of one MDPI paper versus basically every protocol in the influenza world is not acceptable. RDE treated samples should be run and the results included in the manuscript.

There are several published protocols in various journals in which RDE treatment is not strictly performed for virus microneutralization assays

(<https://pmc.ncbi.nlm.nih.gov/articles/PMC9298957/>,

<https://www.nature.com/articles/s41467-021-21954-2#Sec11> ,

<https://www.sciencedirect.com/science/article/pii/S0264410X20301419#s0010>,

<https://journals.asm.org/doi/full/10.1128/cvi.00278-15>). While RDE treatment is commonly performed for HI assays, it is not universal for microneutralization assay protocols as we performed. Further, given the already low to undetectable neutralizing activity measured in vaccinated animals without RDE treatment and evaluation of sham control sera which did not show neutralizing activity, any impact of RDE treatment, which is to remove non-specific HA inhibitors that may inflate neutralizing activity, would be marginal and not change the conclusions of our manuscript. Nevertheless, we performed a standard HI assay in which sera was treated with RDE and heat. No neutralizing activity was detected. This has been added as supplemental figure 3 and lines 86-88.

Reviewer #2 (Remarks to the Author):

1. The study constructed replicating mRNA vaccines targeting HA and NP. To validate the expression level of the mRNA intracellularly, the researchers subjected the cell lysates to both non-denaturing and denaturing treatments, followed by analysis using Western blot; however, they did not demonstrate whether HA formed trimers(Supplemental Figure 1) .

Although we did not show production of trimers, our data suggests that repRNA vaccination presents the HA in relevant conformations to elicit protective immunity against a stringent lethal challenge. Further, we have now evaluated sera from vaccinated mice, vaccinated humans and mice that survived 2.3.4.4b infection against a panel of H5 HAs. We found that the antibodies elicited by repHA vaccination had similar binding profiles as those elicited by authentic A/Vietnam or clade 2.3.4.4b H5N1 viral infection. Thus, while we cannot be certain that repRNA vaccination results in production of authentic trimers, our data indicate that repRNA vaccination elicits antibodies of similar binding profiles as those elicited by authentic virus containing trimeric HA.

The authors still have not verified whether the HA form expressed by the self-replicating mRNA vaccine is in the trimeric form. The formation of HA trimers significantly enhances the induced immune response and protective effects. Therefore, I believe it is necessary to verify this using non-denaturing Western blot experiments.

While we agree that we have not verified the form of expression by our repRNA vaccines, evaluating the western blot performed (provided as source data with manuscript) shows a higher weight band of ~120kda that could be higher complexed HA. Confirmation of the identity of this higher band will require further studies.

Further, we found that mice that survived A/Vietnam infection had antibodies that bound clade 2.3.4.4b HA poorly, similar to our repHA-Vietnam vaccinated mice (Figure 3). These surviving mice were exposed to authentic trimeric HA from infectious virus and still generated antibodies that poorly recognized clade 2.3.4.4b HAs. Thus authentic trimeric presentation of clade 1 H5 HA may still not improve the binding of antibodies elicited to clade 2.3.4.4b HA. Ultimately, we feel our principal finding and conclusion that repHA-Bovine conferred single shot protection against lethal H5N1 challenge in mice is supported by our data. The antigenic presentation by repRNA vaccines is the subject of continued study and optimization and will be the subject of future manuscripts. We have added additional wording that approved vaccines may present the antigen in more immunogenic conformations (Line 204-206) which is in addition to our discussed limitation that we did not evaluate actual stockpiled vaccines (Line 202-211).